# Extracellular RNAs in Bacterial Infections: From Emerging Key Players on Host-Pathogen Interactions to Exploitable Biomarkers and Therapeutic Targets

**DOI:** 10.3390/ijms21249634

**Published:** 2020-12-17

**Authors:** Tiago Pita, Joana R. Feliciano, Jorge H. Leitão

**Affiliations:** iBB-Institute for Bioengineering and Biosciences, Departamento de Bioengenharia, Instituto Superior Técnico, Universidade de Lisboa, Av. Rovisco Pais 1, 1049-001 Lisboa, Portugal; tiagopita@tecnico.ulisboa.pt (T.P.); joana.feliciano@tecnico.ulisboa.pt (J.R.F.)

**Keywords:** extracellular RNAs, membrane vesicles, host-pathogen interactions

## Abstract

Non-coding RNAs (ncRNAs) are key regulators of post-transcriptional gene expression in prokaryotic and eukaryotic organisms. These molecules can interact with mRNAs or proteins, affecting a variety of cellular functions. Emerging evidence shows that intra/inter-species and trans-kingdom regulation can also be achieved with exogenous RNAs, which are exported to the extracellular medium, mainly through vesicles. In bacteria, membrane vesicles (MVs) seem to be the more common way of extracellular communication. In several bacterial pathogens, MVs have been described as a delivery system of ncRNAs that upon entry into the host cell, regulate their immune response. The aim of the present work is to review this recently described mode of host-pathogen communication and to foster further research on this topic envisaging their exploitation in the design of novel therapeutic and diagnostic strategies to fight bacterial infections.

## 1. Introduction

Non-coding RNAs (ncRNAs) are ubiquitous gene regulatory molecules across all life domains, although with specific particularities in plants, animals, and microorganisms. In eukaryotes, ncRNAs can control gene expression by different mechanisms, including chromatin architecture, epigenetics, transcription, splicing, editing, and translation [1,2]. ncRNAs can be classified as long ncRNAs (lncRNAs) with more than 200 nucleotides, and small ncRNAs (sncRNAs) with less than 200 nt [3]. The sncRNAs can further be subdivided into microRNAs (miRNAs) and short interfering RNAs (siRNAS), usually ranging 18–22 nt and 20–24 nt long, respectively [1]. miRNAs and siRNAs can be distinguished by their role in the physiology of the eukaryotic cell. miRNAs are involved in endogenous gene expression regulation, while siRNAs are mainly responsible for protecting the cell from exogenous nucleic acid attack. miRNAs and siRNAs regulate gene expression at the post-transcriptional level by binding their target mRNA by base complementary, generally with an inhibitory effect on gene expression. These interactions depend on two major families of proteins, Dicer enzymes that excise them from their precursors, and Ago proteins that support their silencing effector functions [2,4].

miRNAs were first identified in *Caenorhabditis elegans* in 1993, and since then almost 50,000 have been discovered in 271 species [2,4] miRNAs can be generated from an exclusive gene, from clusters that encode several miRNAs and/or proteins, and even from introns [5]. In animals, miRNAs interaction with their targets mostly involve a partial complementarity that initiates with Watson–Crick base pairing of the seed region of the miRNA nucleotides 2–8 to the target mRNA [2]. Although similar to animal miRNAs, plant miRNAs are classified in hairpin-derived small RNAs (hpRNAs) and double-stranded (ds) RNA-derived small RNAs (siRNAs). The complementarity patterns of the functional miRNA/targets in plants also differ from the animal ones, where a near-perfect complementarity is achieved between miRNA and mRNA [2,6].

Originally described in plants, siRNAs were observed as a transgene- and virus-induced silencing phenomenon [7] and are categorized in trans-acting siRNAs (ta-siRNAs), endogenous siRNAs (endo-siRNAs), and exogenous siRNAs (exo-siRNAs). siRNAs are characterized by the formation of a duplex, sense-antisense, similar to cis-ncRNA in bacteria (described below), leading to the target mRNA degradation [2].

lncRNAs are the longer ncRNAs in eukaryotes and the ones that are poorly understood. Based on their location and neighborhood orientation, lncRNAs can be classified as sense/antisense, divergent/convergent, or intronic/intergenic. They can act through several mechanisms such as a direct regulation of gene expression and activity, a decoy to prevent the attachment of transcription factors to specific promoter regions, a guide to orientate proteins to their target locations, and a scaffold to assist with the assembly of pertinent molecular units [8]. lncRNAs can modulate transcription, whether repressing or activating, by sequestering factors including transcription factors, catalytic proteins, subunits of larger chromatin modifying complexes, as well as miRNAs [3]. Frequently, changes in lncRNAs expression leads to a dysregulation of cellular functions such as cell proliferation, induction of angiogenesis, resistance to apoptosis, promotion of metastasis, and evasion of tumor suppressors. Many lncRNAs have been functionally associated with human diseases such as cancer [9].

On the other hand, bacterial non-coding RNAs were first described in *Escherichia coli*, in the 1960s. The majority of bacterial ncRNAs range from 50 to 400 nucleotides long, being capable of driving the fastest transcriptional regulation [10,11,12,13]. As far as it is known, bacterial ncRNAs are mainly located in intergenic regions and represent about 5% of the total number of bacterial genes [12,14]. Bacterial ncRNAs can target multiple molecular structures, being involved in the regulation of diverse cellular processes, including replication, transcription, translation, energetic and general metabolism, peptidoglycan synthesis, and bacterial virulence [10,15]. Bacterial ncRNAs can play a role in cellular metabolism, iron homeostasis, quorum-sensing (QS), stress response, environmental adaptation, as well as in mechanisms related to bacterial pathogenesis [16]. Mainly acting by antisense base-pairing, bacterial ncRNAs can be classified as cis- or trans-encoded, sharing a full or partial complementarity with their mRNA targets, respectively. Due to their nature, trans-encoded ncRNAs are less specific and usually target multiple mRNAs [15,17]. The majority of the bacterial ncRNAs bind the 5′ untranslated region (5′-UTR) of the target mRNA, although interactions with the 3′-UTR and the coding region have been reported. The interaction between ncRNA and mRNA can result in a translation suppression, with or without RNase E-mediated mRNA degradation. There are also cases reporting that this interaction can lead to an activation of gene expression [15,18]. The regulatory function of a bacterial ncRNA can also be impaired by other RNA molecules such as mRNA and ncRNA [19,20]. Bacterial ncRNAs can also interact with proteins, and particularly important are the interactions with RNA chaperones. RNA chaperones play a crucial role in the regulatory mechanisms of some ncRNAs, by stabilizing the RNA molecules and/or mediating the interaction between sRNAs and their targets. Proteins of the Hfq family are the best characterized bacterial RNA chaperones, being present across kingdoms and highly conserved among bacterial genomes [21,22,23,24]. Another recently characterized RNA chaperone is the ProQ, a chaperone that seems to be involved in interactions with more structured ncRNAs [25].

## 2. Extracellular ncRNAs in Eukaryotes: Release Mechanisms and Relevance in Bacterial Infections

siRNAs were the first ncRNAs described to be involved in intercellular interactions, as a way to disturb other cells, to promote cell-cell communication, or as a defense mechanism against exogenous RNAs. However, an increasing panoply of ncRNAs is being found in the extracellular space, including among others, eukaryotic miRNAs, lncRNAs, siRNAs, piwi-interacting RNAs, and bacterial ncRNAs [26,27]. miRNAs have also already been found in diverse extracellular environments such as blood, urine, saliva, and ascitic fluid [26]. These facts may open the door to engineer the use of miRNAs as biomarkers and, in a research perspective, to pave the way for the understanding of the possible intra- and interspecies communication through exogenous miRNAs.

At least five mechanisms of miRNAs release to the extracellular environment have been described: (i) miRNA bound to RNA-protein complexes. A couple of studies have identified the secretion of miRNAs linked to proteins of the Argonaute family, such as Argonaute 2 (Ago2), a type of protein that is associated to RNA-inducing silencing complex (RISC) [28]. RISC is involved in most of miRNAs regulation [29]. In fact, the majority of miRNAs found in human plasma are bounded to Argonaut proteins, but this seems to be mainly related to cell death and not to a selective secretion of miRNAs [30]; (ii) transport via lipid or lipoprotein particles. In addition to being crucial transporters of steroids, triglycerols, cholesterol, and fat-soluble vitamins, low-density lipoproteins (LDL) and mainly high-density lipoproteins (HDL) can also play a role in miRNAs intercellular communication [31,32]. The loading mechanism seems to involve divalent cation bridging between miRNAs and HDL [31]; (iii) inside microvesicles. Microvesicles are formed by plasma membrane by budding or fission, and therefore, their lipid content is quite similar to the parent cell membrane [33]. Microvesicles can be a way of secretion of many types of molecules, including nucleic acids, being also responsible for the cell-to-cell communication by miRNAs exportation in several clinical conditions [34]. Although there is evidence of the involvement of microvesicles miRNAs on intracellular communication, their relevance in infection conditions remains unknown, as well as the sorting and loading processes [35]; (iv) inside vesicles from apoptotic bodies. Apoptosis is a natural process of controlled cell death by eukaryotic cells. In this process, the release of apoptotic bodies, the greatest vesicles secreted by eukaryotic cells, is common [36]. As with microvesicles, the content can be vast, including miRNAs, mRNAs, and DNA fragments, but it can also be selective, and under specific conditions, some miRNAs can be highly represented on these vesicles [37]. Again, the sorting mechanism remains unknown [35]; (v) inside exosomes. Despite their origin or structure, extracellular vesicles (EV) seem to be the most usual mechanism to selectively export ncRNAs to the extracellular space [27]. Exosomes seem to play a special role in cell-to-cell communication on infection conditions [35]. Exosomes are generated inside endosomes or multivesicular bodies (MVBs) and released through fusion of these exosome-enriched late endosomes with the plasma membrane (Figure 1) [38]. Although the process of sorting and loading of ncRNAs is still poorly characterized, is about exosomes that we know the most. In animals, the heterogeneous nuclear ribonucleoprotein A2B1 (hnRNPA2B1) was described to control the exosomal loading of miRNAs by binding to specific “EXOmotifs” on these miRNAs [39]. The most detailed example found in the literature is about colorectal cancer cells. In this case, the KRAS-MEK signaling pathway seems to be responsible for the regulation of exosomal loading of the RISC component Argonaute 2, an RNA-binding protein and a key effector of miRNA-guided RNA silencing process [40]. In addition, during the sorting and loading of ncRNAs in exosomes, other RNA-binding proteins are also suggested to participate, such as the Y-box protein required for the miR-233 secretion by human embryonic kidney (HEK)293T cells [41], and the SYNCRIP protein required for miRNA sorting in hepatocytes. miRNAs found in those exosomes possess an extra-seed sequence (hEXO motif) that binds to these RNA-binding proteins [42]. Still, in the hepatic system, the RNA-binding protein Vps4A was found to mediate the flux of miRNAs through exosomes. Vps4A facilitates the secretion of oncogenic miRNAs in exosomes while promoting the accumulation and uptake of tumor suppressor miRNAs in cells. A downregulation of this protein was observed in hepatocellular carcinoma (HCC) cells [43]. In addition, the transcriptional regulation of miRNAs expression or of their targets also implies a miRNA sorting regulation on exosome secretion, as shown in macrophages and endothelial cell communication [44]. The mechanism of sorting and loading of miRNAs in infection conditions needs to be investigated.

In the context of infectious processes, miRNAs have been increasingly implicated in the eukaryotic response to viruses, nematodes, and bacterial pathogens [47]. Specific miRNAs, such as miR-155, miR-146a, miR-21, and the let-7 family of miRNAs have been demonstrated to be involved in the regulation of the immune response to infection (the mechanisms involved were recently reviewed by Kumar et al. [48]. Immune cells are able to release microRNA-containing exosomes that can be uptake by recipient cells. During antigen recognition, when antigen-specific T cells form an immunologic synapse with antigen-presenting cells (APCs), miRNAs are unidirectionally transferred between cells by exosomes [49]. More recently, increasing evidence has also suggested that infections with pathogenic organisms lead to significant changes in the miRNA exosome content (miRNA abundance and profile) [50,51]. Some of these exosome-delivered miRNAs seem to immunomodulate the inflammatory response, however, their specific role in host vesicles derived from bacterially infected host cells is not completely understood. For instance, higher levels of miR-18a, a miRNA that promotes the intracellular *Mycobacterium tuberculosis* survival by counteracting autophagy, were detected in macrophages infected by *M. tuberculosis* and in their derived exosomes [52]. However, the miR-18a impact in exosome-receiving cells remains unclear. On the other hand, miR-155, a prototype multifunctional miRNA that exhibits crucial roles during innate or adaptive immune responses, was shown to be loaded in exosomes derived from *Helicobacter pylori*-infected macrophages. This miRNA exacerbates inflammatory responses in recipient macrophages by promoting the expression of inflammatory cytokines, such as TNF-α, IL-6, and IL-23, which help to inhibit the proliferation of *H. pylori* [53]. Exosomes containing miR-146a and miR-155 were also described to be secreted by murine bone marrow-derived dendritic cells (BMDC) after exposure to LPS. Both miRNAs seem also to be efficiently transferred to recipient cells, modulating the expression of inflammatory genes and cell responses to endotoxins [54].

## 3. Extracellular ncRNAs in Bacteria

Communication between cells is not an exclusive phenomenon in animals, or even eukaryotes, and neither is just an intraspecies phenomenon. This communication can also occur among bacteria and across kingdoms, as in eukaryotes. Although quorum-sensing is arguably the best-known communication systems in prokaryotes, ncRNAs also play a fundamental role in communication among prokaryotes. In bacteria, the standard mechanism of ncRNAs secretion is through extracellular vesicles, also mentioned as membrane vesicles (MVs), vehicles that can be used either for the secretion of both non-soluble and soluble particles [55]. Typically, bacterial MVs are 50–250 nm spheroid particles derived from the bacterial membrane. The production of these vesicles is strongly dependent on the environmental conditions and the bacterial growth phase [56,57,58]. In addition to helping in cell-to-cell communication [59], MVs have been recognized for their role in the acquisition of nutrients [55], stress responses, delivery of toxins, and evasion of the host defense system [60]. Both Gram-positive and Gram-negative bacteria are known to release vesicles, although the mechanisms involved in the biogenesis of these MVs differ among these organisms [45,61,62]. Due to the composition of the Gram-negative cell wall, membrane vesicles secreted by these bacteria are commonly designated as outer membrane vesicles (OMVs), and these are the best-studied so far.

### 3.1. OMVs Biogenesis

It is well known that the cell wall of Gram-negative bacteria comprises two membranes (inner and outer membrane), separated by the periplasm containing a thin peptidoglycan layer. These two membranes are constitutively different concerning their lipid and protein composition, being the lipopolysaccharide (LPS) the main component of the outer membrane. The periplasmic space is a viscous area, holding the peptidoglycan layer [63]. The cell wall of Gram-negative bacteria is vital for cell maintenance, playing various roles, including nutrient acquisition, adherence, secretion, signaling, and protection from the environment [55]. The OMVs are the product of periplasm secretions encapsulated in the outer membrane, in a process resembling budding in yeast, as depicted in Figure 1. Although easily understandable, the biogenesis of OMVs is still misunderstood, and so far, no definitive model was established. Four major models for OMV genesis have been proposed: (i) Loss or relocation of covalent linkages between the outer membrane and the underlying peptidoglycan layer via lipoproteins or other components. When the defect occurs, the faster growth rate of the outer membrane compared to that of the underlying cell wall allows the outer membrane to protrude and finally to generate the OMV [64]; (ii) The interaction between the outer membrane and turgor pressure, which is generated from the accumulation of peptidoglycan fragments or misfolded proteins in the periplasmic space, causing the outer membrane to bulge and finally to pinch off [65]; (iii) Increasing the amount of membrane curvature-inducing molecules, such as B-band LPS and Pseudomonas quinolone signal (PQS) that can enhance anionic repulsions between LPS molecules, resulting in membrane blebbing by sequestering divalent cations. The asymmetric expansion of the outer leaflet of the outer membrane leads to the formation of OMVs. Although well accepted and one of the most studied, this model only applies to *Pseudomonas aeruginosa* [66]; (iv) Phospholipid accumulation in the outer leaflet of the outer membrane, regulated by the expression of *vacJ* and/or *yrb* genes, resulting in an asymmetric expansion of the outer leaflet and subsequent promotion of an outward bulging of the outer membrane [67]. Although this process does not overlap the other models, it can be in fact a complement to explain the OMV genesis. Summarizing, the OMV biogenesis seems to rely on three main mechanisms: (1) dissociation of the outer membrane in specific zones lacking proper attachments to underlying structures (e.g., peptidoglycan); (2) the presence of misfolded proteins, which accumulates in nano-territories where crosslinks between peptidoglycan and other components of bacterial envelope are either locally depleted or displaced; (3) specific changes in LPS composition that modulate OMV biogenesis, presumably by generating a differential curvature, fluidity, and/or charge in the outer membrane [68].

### 3.2. OMVs Composition and RNA Secretion

In addition to membrane components, OMVs are often composed of proteins of the inner membrane, cytoplasmic proteins, DNA, RNA, ions, metabolites, and signaling molecules [67,69]. OMVs play an important role in the physiology and pathogenesis of Gram-negative bacteria, requiring a significant amount of energy to be produced. Horizontal gene transfer, biofilm formation, intra- and interspecies communication, stress response, delivery of toxins and other biomolecules, resistance to antibiotics, killing of competing microbial cells, adherence to host cells, complement absorption, and immunomodulation [67] are some of the functions that have been attributed to OMVs. The potential therapeutic use of OMVs to fight some bacterial infections is also an interesting topic of research [70].

Accumulating evidence has suggested that OMVs’ content comprises intracellular components such as DNA and RNA. However, the specific roles of OMV-nucleic acids, and how this content is sorted to OMVs remains unclear. Although some cases of free ncRNAs outside the bacterial cell have been reported, as far as we know, most of the research performed involves RNAs encapsulated in OMVs. Considering their unstable nature, more prone to RNAse degradation, it seems more plausible that bacterial ncRNAs are proposedly secreted associated with proteins, inside vesicles, or even with an RNAse-resistant extensive secondary structure, as observed in eukaryotes and virus [71,72]. In addition, to represent the protection required for the RNA content, OMVs are also an easy, precise, and specific delivery system [73]. Despite all this, the secretion of bacterial ncRNAs remains to be elucidated, as well as the non-selective vs. specific enrichment of individual transcripts during the vesiculation process of bacterial RNA [73].

Koeppen et al. observed that microRNA-sized small RNA (msRNA) seems to be overrepresented in *P. aeruginosa* OMVs compared to the intracellular content [74]. First proposed by Lee and Hong (2011), msRNAs were later characterized for the first time in *E. coli* as a new kind of bacterial RNA containing 15–28 nt in length, and proposed to play a regulatory role similar to that of the eukaryotic miRNAs [75,76]. In addition to an enrichment on RNAs with a medium sequence length of 24 nt, Koeppen and co-authors also proposed a selective packaging of ncRNAs in *P. aeruginosa* OMVs containing an RNA profile distinct from that of the *P. aeruginosa* cells [74]. In a study performed by Ghosal et al., an enrichment in short RNAs (15–40 nt) was also found on the secreted content of *E. coli* OMVs, with differences in the profiles of intracellular, OMV-associated, and OMV-free extracellular RNA [77]. Despite all of this, it is still speculative whether the RNA packaging inside MVs is necessary. Secreted RNAs were also described for other bacteria such as *Streptococcus* sp., *Vibrio cholerae*, *Porphyromonas gingivalis*, and *Helicobacter pylori* [78,79].

Recently, Pagliuso et al. have demonstrated that RNA-binding proteins (RBPs) can play a role in RNA secretion. These authors found that the bacterial pathogen *Listeria monocytogenes* secretes a small RBP, Zea, which was shown to bind and export RNAs to the extracellular medium [80]. In addition, a linkage between Hfq and the bacterial membrane was recently evidenced, suggesting a possible new role for this RNA chaperone in exporting sRNA to the outside of the bacterial cell [21]. The Sec2A secretory system of *Mycobacterium tuberculosis* was also found to be involved in RNA secretion [81].

Although the role of bacterial secreted RNAs (seRNAs) remains scarcely explored and known, research carried out during the last decade suggests that these RNAs are multipurpose molecules. So far, it is known that seRNAs can be involved in diverse cellular processes across living beings, establishing regulatory networks between the producing bacteria and other bacteria, intra or interspecies, and with host cells in the case of pathogens [78]. In the next section, examples of bacterial secreted RNAs that mediate communication are presented.

### 3.3. Specific Examples of Bacterial Secreted Micro-Size RNAs

The secretion of RNA to the extracellular space seems to be a common feature among nematodes, plants, bacteria and fungi, as well as mammals [82]. Being a common feature among so different organisms, most likely the general purpose should be the also common, communication. In the human body, RNAs (sRNAs, ncRNAs, lnRNAs) from distinct origins are circulating in fluids such as blood. Similarly, the presence of bacterial seRNAs has also been described in human and murine plasma and serum, derived from diet and/or the microbiome [83]. There is also evidence that a functional delivery of RNA occurs between eukaryotic cells [84], and the presence of bacterial seRNA (most of them from OMVs) in host cells has also been reported for an increasing number of organisms [74,79]. The delivery of these seRNAs in vivo seems to be highly dependent on the bacterial lifestyle (extracellular, intracellular, or facultative).

The first allusion to seRNAs goes back to 1989, when Dorward, Garon, and Judd speculated about RNA secretion by *Neisseria gonorrhoeae*. However, the first full characterization of seRNAs was performed in *E. coli* by Ghosal et al. in 2015 [77,85]. Since then, several descriptions of extracellular RNA have been carried out for some bacteria, which will be explored in further detail below.

#### 3.3.1. *Escherichia coli*

*E. coli*, as a model of Gram-negative bacteria, was one of the first bacteria used to investigate the presence of secreted RNAs. Ghosal et al. used the model bacterium *E. coli* K-12 sub-strain MG1655 to investigate the RNA content of OMVs, as well as their free form on the extracellular space [77]. The extracellular RNA size ranges between 15 and 40 nucleotides and these molecules derived from specific intracellular RNA species. When compared, distinct RNA expression profiles were also observed among OMV, OMV-free, and the intracellular medium. The extracellular RNA is enriched in specific cleavage products of functionally important structural non-coding RNAs, including tRNAs, 4.5S RNA, 6S RNA, and tmRNA (transfer-messenger RNA). However, no concrete information was revealed by the authors about the possible purpose of the bacterium for secreting these RNAs [77]. Using the uropathogenic *E. coli* (UPEC) strain 536, Blenkiron et al. evaluated the ability of RNA to reach the intracellular space of epithelial cells. About 1% of OMVs RNA cargo was delivered into cultured bladder epithelial cells, reaching the cytoplasm and nucleus. The internalization in the epithelial cells of the *E. coli* non-coding RNA CsrC, an ncRNA involved in carbon storage, was also confirmed by RT-PCR [86,87].

#### 3.3.2. *Pseudomonas aeruginosa*

The Gram-negative bacterium *P. aeruginosa* is a ubiquitously distributed opportunistic pathogen affecting patients with immunosuppressive and chronic conditions, such as chronic obstructive pulmonary disease and cystic fibrosis. In these patients, *P. aeruginosa* infections are difficult to treat due to several antibiotic resistance mechanisms and the propensity of these organisms to form multicellular biofilms [88].

Similar to *E. coli*, sRNAs with a range size of 15 to 45 nt long were also found as selectively exported in *P. aeruginosa* OMVs [74]. Among them, the sRNA52320 stands out, since it is a seRNA with predicted mRNA targets in host cells that is transported by OMVs to be delivered on primary human bronchial epithelial (HBE) cells. In addition to being the first OMV-associated sRNA characterized in *P. aeruginosa*, for the first time in any bacteria, Koeppen and co-authors provided evidence of a bacterial seRNA playing a role in host cells [74]. Being a fragment of a *P. aeruginosa* methionine tRNA, sRNA52320 was found in high abundance in OMVs and impacts the immune response of host cells. sRNA52320 can reduce the LPS-induced as well as OMV-induced interleukin 8 (IL-8) secretion by cultured HBE cells. This seRNA is also involved in the attenuation of OMV-induced KC cytokine secretion and neutrophil infiltration in mouse lung [74].

Turnbull et al. also detected RNA in MVs derived for explosive cell lysis in *P. aeruginosa* PAO1. Intact ribosomal RNAs, such as 16S and 23S, were among the identified RNAs, which indicates the protective role of MVs from ribonucleases. The content of those MVs revealed an enrichment of mRNAs that are typically expressed as part of the SOS response in *P. aeruginosa* following exposure to oxidative stress, DNA-damaging agents, or antibiotics. This SOS response is observed on a small percentage of a population grown under standard growth conditions. The authors also identified sRNAs on MVs, such as PA3305.1 (PhrS), which was found to be less abundant on MVs than on planktonic cells [89]. Since PhrS stimulate Pseudomonas quinolone signal (PQS) production, the reduced cargo on MVs seems to corroborate the thesis of an explosive cell lysis-mediated MV production independent of PQS [90].

#### 3.3.3. *Helicobacter pylori*

*H. pylori* is a Gram-negative, microaerophilic, human pathogen that specifically colonizes the gastric mucosa, causing persistent infections associated with the induction and progression of several gastric disorders. *H. pylori* constitutively releases OMVs from its outer membrane [91]. In addition to carrying a plethora of virulence factors, *Helicobacter* OMVs play an important role in the persistence of *H. pylori* infections and promote biofilm formation [91,92,93]. Through RNAseq analysis, more than a half of million unique small noncoding RNA sequences, with a size of 15 to 55 nucleotides, were identified in OMVs purified from the supernatants of *H. pylori* strain J99 [79]. In addition, using quantitative RT-PCR and microscopy, the authors suggested that the transfection of seRNAs through *H. pylori* OMVs to cultured human gastric adenocarcinoma (AGS) cells occurs. Moreover, two seRNAs, sR-2509025 and sR-989262, were found to be enriched in OMVs, and were reported to have a functional effect in reducing the lipopolysaccharide stimulation of IL-8 secretion in cultured AGS cells [79]. Similar to the findings on *P. aeruginosa* by Koeppen et al. [74], these data suggest that sRNAs within *H. pylori* OMVs might play a fundamental role in directly tuning the host immune response [79].

#### 3.3.4. *Listeria monocytogenes*

*L. monocytogenes*, the causative agent of listeriosis, is a ubiquitously occurring facultative intracellular Gram-positive bacterium. A variety of phagocytic and nonphagocytic host cells can be infected by *L. monocytogenes*, which can replicate intracellularly and spread from cell-to-cell [94]. Considering their lifestyle, these bacteria can produce MVs both in vitro and in vivo [95]. A vast repertoire of seRNAs was also identified, mainly represented by ncRNAs. The ncRNAs secreted by *L. monocytogenes* revealed the ability to induce type I interferons (IFNs) [96]. The induction of the IFN response has been suggested to increase host susceptibility to *L. monocytogenes*, as well as to other intracellular pathogens [97]. The ncRNA rli32, which is highly conserved among *L. monocytogenes* strains, was shown to be a potent inducer of IFN-β expression. This response was also revealed to be RIG-I (retinoic acid inducible gene I)-dependent, and to trigger an inhibition of influenza virus replication. Moreover, rli32 seems to play a major role in the ability of *L. monocytogenes* to grow inside host cells. rli32 overproduction promotes intracellular bacterial growth, while its deletion restricted bacterial growth [98]. Pagliuso et al. upgraded the present knowledge on seRNAs host cell regulation by revealing that Zea, the previously mentioned RNA-binding protein, may be responsible for *L. monocytogenes* RNAs accumulation in the extracellular medium. Zea has the ability to binds to a group of ncRNAs, including RIG-I [80], and can be involved in rli32 RIG-I dependent induction of IFN-β expression, a hypothesis that remains to be investigated. *L. monocytogenes* seRNAs research represents a major step in our knowledge on bacterial seRNAs, especially among Gram-positive bacteria.

#### 3.3.5. *Salmonella* sp.

*Salmonella* includes a group of enteric Gram-negative bacteria that can cause food-borne infections (salmonellosis) in humans. In addition to being facultative anaerobes, *Salmonella* species are also facultative intracellular pathogens. When grown on a medium mimicking host-pathogen interaction, *Salmonella enterica* serovar Typhimurium revealed a selective secretion of seRNAs in OMVs [69]. The study also evidenced what seems to be a specific processing or degradation process, since while some transcripts maintained the same reads coverage profile in OMVs, others presented a different pattern. It was also demonstrated that *Salmonella* RNAs are protected by OMVs, since full-length transcripts (SsrS, CsrC, 10Sa, and rnpB) present in OMVs remained intact after a digestion essay [69]. Gu et al. have shown that once inside intestinal epithelial HT-29 cells, *Salmonella enteritidis* can secrete RNAs [99]. Moreover, those RNAs can even just assume their mature form after processing by the host cell machinery to assume microRNA-like fragment structures. This is the case of Sal-1, which results from the processing of the *Salmonella* 5′-leader ribosomal RNA transcript. The processing of Sal-1 precursors to mature Sal-1 is dependent on the host cell Argonaute 2 (AGO2) but not on Dicer. These authors also showed a reduced resistance of *Salmonella* to host defense by depleting Sal-1, disclosing a novel strategy used by *Salmonella* to evade the host immune clearance [99].

#### 3.3.6. *Mycobacterium tuberculosis*

Mycobacteria are small rod-shaped bacilli that can cause a variety of diseases in humans. *Mycobacterium tuberculosis* is the causative agent of tuberculosis, and a facultative intracellular pathogen that can persist within the host [100]. *M. tuberculosis* has been described to secrete small RNAs inside host cells such as murine macrophages, ranging in size from 15 to 43 nt [101]. Obregón-Henao et al. hypothesized that the seRNAs could be responsible for the host cell apoptosis induced by *M. tuberculosis*. In fact, these authors found a linkage between seRNAs and induced apoptosis, describing that RNA fragments compromise the ability of human monocytes to control *M. tuberculosis* infection through a caspase-8 -dependent, caspase-1 and TNF-α -independent pathway [102]. Moreover, Cheng et al. observed that RNA is delivered into macrophages by EVs that depend on the *M. tuberculosis* SecA2 secretory system. In addition, they also showed that EVs released from *M. tuberculosis*-infected macrophages stimulate a host RIG-I/MAVS/TBK1/IRF3 RNA sensing pathway, leading to type I interferon production in recipient cells. This mechanism induces an immune response by host cells, increasing trafficking of *M. tuberculosis* into LC3- and Lamp-1-positive vesicles, enhancing bacterial killing. The treatment of *M. tuberculosis*-infected macrophages or mice with a combination of moxifloxacin and EVs isolated from *M. tuberculosis*-infected macrophages significantly lowered bacterial burden relative to either treatment alone. These findings pave the way for the exploitation of EVs containing bacterial secreted RNAs into medical therapies [81].

#### 3.3.7. Periodontal Pathogens: *Aggregatibacter actinomycetemcomitans*, *Porphyromonas gingivalis*, *Treponema denticola* and *Streptococcus sanguinis*

A study on sRNAs present in OMVs from the periodontal pathogens *Aggregatibacter actinomycetemcomitans*, *Porphyromonas gingivalis*, and *Treponema denticola* showed that, despite the different features, these three bacteria exhibit a similar pattern in terms of seRNAs, as it has been described for other Gram-negative bacteria [103]. msRNAs were found in OMVs, which therefore can penetrate host eukaryotic fibroblast cells (NIH3T3). However, the release of msRNAs inside host cells remains to be demonstrated. A combination of synthetic msRNA oligos, one from each mentioned periodontal pathogen, was transferred into Jurkat T cells and a decrease of the anti-inflammatory cytokines Interleukin (IL)–5, IL-13, and IL-15 was observed [103]. Further analyses are necessary to clarify the function of each msRNA. The work of Ho et al. (2015) corroborated the ability of *P. gingivalis* vesicles to invade host cells, showing that 70–90% of human primary oral epithelial cells, gingival fibroblasts, and human umbilical vein endothelial cells carried vesicles from *P. gingivalis* 33,277 after exposure for 1 h. Moreover, *P. gingivalis* vesicles, containing both DNA and RNA molecules, were found to be capable to transfer DNA between *P. gingivalis* strains, also displaying an inhibitory effect in the *Streptococcus gordonii* biofilm formation ability. These results evidence a major role of *P. gingivalis* vesicles on bacterial survival in the oral cavity, as well as on the induction of periodontitis, although the involvement of seRNAs on such regulatory mechanism remains to be proved [104]. Furthermore, Han et al. provided evidence for the cytoplasmic delivery and activity of *A. actinomycetemcomitans* OMV-derived small seRNAs in host gene regulation, finding an enhanced TNF-α production via TLR-8 and NF-κB signaling pathways [105]. In another study, Choi et al. reported the detection of msRNAs from the Gram-positive oral pathogen *Streptococcus sanguinis* in the host intracellular medium, being two of them also present on the *S. sanguinis* MVs [106]. Figure 2 illustrates the general processes of host cell immune response to EVs and its internal RNA content.

#### 3.3.8. Additional Examples

Despite the limited number of studies, bacterial seRNAs have been described for a few more organisms. For instance, Sjöström et al. reported the presence of ncRNAs in OMVs of *Vibrio cholerae* O1 El Tor strain, mainly originated from intergenic regions [107]. More recently, the OMV-associated sRNA SsrA was shown to be crucial for the symbiotic interactions between *Vibrio fischeri* and the squid *Euprymna scolopes* in the light organ. SsrA, a small stable bacterial RNA molecule that is present in OMVs, can be found in the hemolymph of the adult squid. In the absence of SsrA, the partnership between these two organisms was compromised: while *V*. *fischeri* cells were not able to persist in the light organ, the host robustness was reduced and a heightened in immune response seems to occur. This example demonstrates the potential for sRNA molecules to be key elements in the language of beneficial host-microbe associations [108]. Furuse et al. used deep sequencing to discover microRNA-like fragments that could be expressed by intracellular bacterial pathogens such as *Legionella pneumophila*, *Chlamydia trachomatis*, and three mycobacterial species *Mycobacterium marinum*, *M. smegmatis*, and *M. tuberculosis.* The authors searched for small RNAs associated with the cellular RNA-induced silencing complex RISC. In the case of *M. marinum*, the small RNA MM-H was found. This small RNA has the features of a microRNA of bacterial origin, including a defined pre-microRNA stem-loop, a size close to 22 nt long and an efficient effect on the target mRNA when the sRNA is properly expressed. It was also demonstrated that a host mRNA-target, the MM-H-specific RLuc reporter, is repressed by the in vitro MM-H overexpression. When infecting host cells, all the bacterial species analyzed revealed the ability to secrete ncRNAs, many of them msRNAs that were found to bind to RISC [101,109]. In a rare description of RNA secretion on MVs by Gram-positive bacteria, Resch et al. showed that group A *Streptococcus* (GAS) also export RNAs through MVs. However, different from most of the descriptions for Gram-negative bacteria, GAS membrane vesicles seem to be enriched in intragenic RNA instead of intergenic RNA, such as tRNAs and small noncoding RNA species, as seen for example in *E. coli* [77,110]. Another interesting case is the OMVs produced by *Borrelia burgdorferi*, which are enriched on plasmid-encoded transcripts compared to the intracellular content. The OMV-enriched transcripts were associated to biological processes related to DNA integration and recombination, suggesting that OMVs released by *B. burgdorferi* may contain messengers involved in genomic rearrangement, one of the biological activities known to be mediated by these vesicles. Most ncRNAs found in MVs are derived from chromosomes and are more abundant in vesicles [111]. A selective RNA secretion was also described to be required for the in vitro formation of biofilms by non-typeable *Haemophilus influenzae*. It was also suggested that extracellular RNA is required for the initial attachment of the biofilm, but not for its maintenance [112]. The presence of RNA on OMVs was also described for marine bacteria [113] and more recently for *Staphylococcus aureus* [114]. Table 1 summarizes the current knowledge on seRNA from bacterial species.

## 4. seRNAs in Novel Diagnostic and Therapeutic Approaches

Since microbial seRNAs can be found circulating within the human body, one can speculate about their exploitation to diagnose human infections, by using these seRNAs as biomarkers. Actually, the research carried out so far has proved that this hypothesis is not just possible, but also very promising. Technically, it would require methodologies similar to the ones developed for the detection of SARS Cov-2 viral RNA in human samples, the molecular diagnostic technique used worldwide in the COVID-19 pandemics that is afflicting all of us.

Features of seRNAs that can potentiate their use as biomarkers, especially of those resembling miRNAs, include key regulatory functions and dysregulation in disease, cell type- and tissue-specific expression patterns, their relative ease of sampling and measurement, and their stability in extracellular environments [82]. The most advanced research in human diagnosis by circulating non-coding RNAs is related to cancer. The secretion of ncRNAs by human healthy or cancer cells have been proven to play a significant role in cell to cell communication and these molecules can be detected in body fluids such as blood and urine [27]. The potential of circulating miRNAs as biomarkers of disease has been demonstrated for various types of cancer. Still, despite the vast literature regarding circulating ncRNAs as biomarkers of cancer, the lack of high specificity and the low levels of reproducibility have restricted clinical applications to a single ncRNA, the urinary PCA3, already approved for molecular diagnosis [115]. Extracellular miRNAs can also be exploited to design novel diagnostic/prognostic biomarkers of infectious disease. The release of miRNAs by host cells has been demonstrated in several infection cases such as human tuberculosis (*M. tuberculosis*), sepsis caused by multiple infectious agents, and viral hepatitis [116]. Fecal miRNAs are also promising as a new way of diagnosis. The discovery of a different pattern of secreted miRNAs in disease conditions such as colorectal cancer and celiac disease, as well as the use of miRNAs to modulated gut microbiome metabolism, have paved the way for the development of new biomarkers based on these secreted molecules [117]. In the field of animal health, a total of 14 secreted miRNAs were found to be differentially expressed in bovine milk exosomes in response to a *Staphylococcus aureus* infection in the mammary gland [118]. This example gives evidence that a selective secretion of ncRNAs can occur in an infection condition, and biomarkers of infections can be originated from different sources, by the infectious agent itself, or by host cells, as a mechanism of defense. The potential of using seRNAs as infections biomarker exists, however, no real application has become available so far.

Although the research of circulating ncRNA is especially focused on human-derived ncRNA, the origins of the RNAs present in human body fluids is by far more extensive. Almost all kind of RNAs are found in blood and extracellular vesicles: mRNA, rRNA, tRNA, miRNA, piRNA, snoRNA, vault RNA, Y RNA, including RNAs from different sources of tissues and other organisms, arising from human skin, gut microbiota, (including viral microbiota) and also from the diet [119]. Small RNAs from bacteria, fungi, and plants can be detected in human plasma, mostly originating from the human microbiota, but also likely to come from diet [83,120,121]. Still, the efficacy and accuracy of a diagnosis based on bacterial secreted RNAs present in human blood plasma remain to be proven.

The human gut microbiome, for instance, can be related to pathophysiology of bowel disorders, obesity, atherosclerosis, diabetes, rheumatoid arthritis, and neurodevelopmental disorders [120]. Moreover, the presence of plant or bacterial RNAs in human circulatory systems was also correlated to localized breakdown of the epithelial barrier in early stages of intestinal inflammatory and gastrointestinal diseases, such as celiac disease, colorectal cancer, food allergies, inflammatory bowel disease, type 1 diabetes, and others [82]. These facts, together with the easiness of obtaining blood samples, highlight the potential of using extracellular bacterial RNAs as biomarkers on human health care.

The idea of using RNA interference for therapeutic use is not new, despite the fact that patisiran, the first RNAi-based drug, was only approved in 2018 [122]. Additionally, Miravirsen, the first miRNA-targeted drug, is still under clinical validation, currently in Phase II clinical trial [117]. There is a growing hope in the use of these kinds of therapies, but the challenges associated with safety and efficiency are huge [122]. The use of bacterial OMVs as vehicles for drug delivery has also been in the highlights, including gene therapy. However, studies on the use of OMVs for human therapies remain scarce [123]. OMVs also have been studied for vaccine usage [70].

Targeting bacterial ribosomal RNA is nowadays a common therapeutic measure, based on the use of antibiotics such as streptomycin, spectinomycin, tetracycline, lincomycin, clindamycin, and chloramphenicol [124]. However, the emergence of antibiotic resistance is turning these drugs obsolete. Effective drugs targeting other bacterial RNA remains just a promise [125]. Moreover, regarding therapies including secreted RNAs, just searching about it is a challenge, suggesting the lack of knowledge on this topic.

## 5. Conclusions

RNA secretion is a ubiquitous strategy used by a great number of organisms, from bacteria to mammals, all with a common purpose, communication. Communication plays a central role in living organisms, whatever the intention is cooperation or invasion. In human body fluids almost all kinds of RNA can be found, coding and non-coding RNAs, originating from the organism’s own cells, from diet and microbiome, and from different backgrounds, including virus, bacteria, fungi, and parasites. To establish communication with host cells and its peers, bacteria evolved several ways to secrete RNAs into the extracellular environment. However, as in eukaryotic cells, the preferred mechanism seems to involve secretion inside vesicles, providing protection from RNase degradation. Data from different bacteria strongly indicate that seRNAs tend to be small non-coding RNAs, and that its secretion is selective. However, the mechanisms of RNA cargo and sorting remains to be fully understood. The effects of bacterial seRNAs on host cells is starting to be disclosed, with a few examples showing that some non-coding RNAs can reach host cells and impact their biology. In addition, examples are available of bacterial seRNAs that can be processed within host cells and manipulate their immune response, leading to an enhanced ability for invasion and colonization of those cells.

Although promising, the exploitation of seRNAs in the design of novel diagnostic or therapeutic approaches is still far from reality. The challenges are huge and further advances in basic knowledge on seRNAs are required. Several questions are still unsolved: How are seRNAs sorted and secreted? What is their impact on microbial communities and on interactions with the host? How can these RNA molecules be targeted? How can they be exploited envisaging the development of novel and robust methods for human disease diagnosis? And what about RNA secretion by the large number of microorganisms that is still to be explored? The potential of this field of knowledge is huge and is just waiting to be explored.

## Figures and Tables

**Figure 1 ijms-21-09634-f001:**
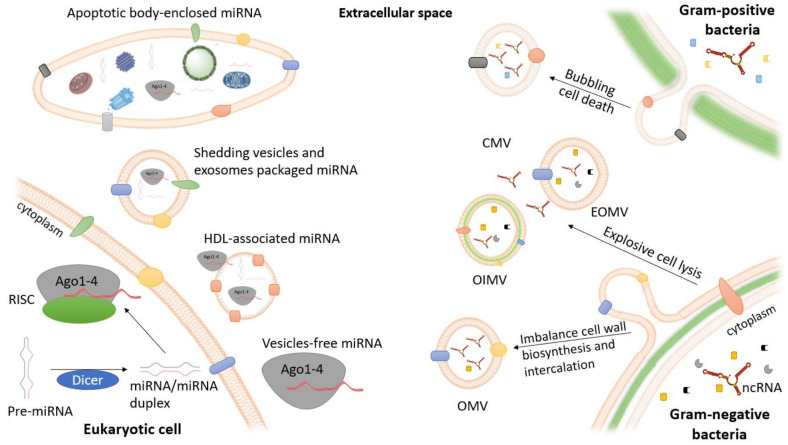
Modes of extracellular miRNA packaging in eukaryotic cells and routes leading to the formation of different membrane vesicle types in bacteria. Extracellular miRNAs can be cargo in membranous vesicles or can be vesicle free and associated with either Argonaute (Ago) proteins alone or incorporated into HDL particles. Apoptotic bodies, shedding vesicles, and exosomes are three types of membranous vesicles that contain these extracellular miRNAs. Apoptotic bodies can also contain various cellular organelles including mitochondria and nucleic acids [28]. In bacteria, distinct membrane vesicles are formed by Gram-negative and Gram-positive bacteria. Blebbing of the outer membrane and explosive cell lysis are the two main routes for vesicle formation in Gram-negative bacteria. The membrane vesicles from these bacteria can be divided into outer-inner membrane vesicles (OIMVs), explosive outer membrane vesicles (EOMVs), and traditional OMVs according to their formation routes, structures, and compositions [45]. In Gram-positive bacteria, membrane vesicles are formed by a mechanism involving the prophage-encoded endolysin that generates holes in the peptidoglycan cell wall, allowing the cytoplasmic membrane material to protrude into the extracellular space and release the cytoplasmic membrane vesicles (CMVs). CMVs can contain membrane and cytoplasmic components [46].

**Figure 2 ijms-21-09634-f002:**
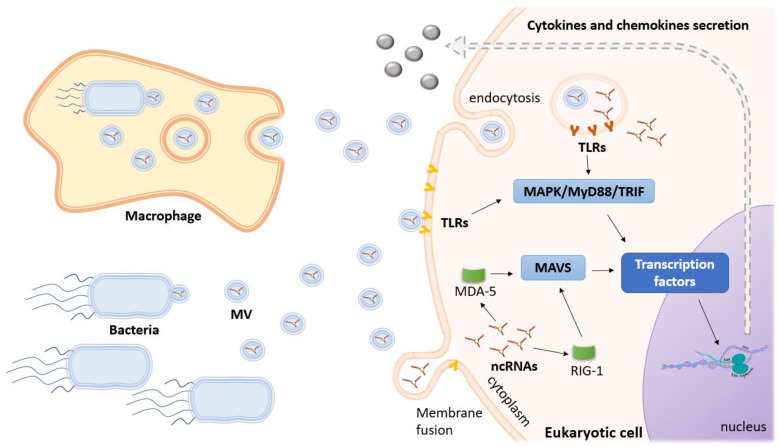
Activation of innate immune receptors by bacterial membrane vesicles and their RNA content. When bacterial EVs released by extracellular or intracellular bacteria reach the target cells, the sensing of these vesicles and/or its RNA content could occur at different intracellular locations. The pathogen-associated molecular patterns (PAMP) on the outside of these vesicles can induce the host innate immune response by activating the toll-like receptor and MAP-kinase (TLR/MAPK) signaling pathway, leading to the activation of transcription factors that induce the production of inflammatory mediators. The RNA cargo of EVs internalized by endocytosis can also be released and sensed by endosomal receptors. These receptors signal through the adaptor molecules MyD88 or TRIF, resulting in the activation and nuclear translocation of transcription factors that will induce production of inflammatory mediators and type I interferons. RNAs delivered into the cytoplasm after fusion of EVs with the host cell plasma membrane might be sensed by cytoplasmic RNA sensors such as RIG-I and MDA-5. Engagement of these receptors triggers signaling through the adaptor protein MAVS, which signals to induce the production of inflammatory mediators and type I interferon. On the other hand, some bacterial sRNAs delivered into receptor cells can target mRNAs from components of the mentioned signaling pathways, leading to the reduction of the inflammatory mediators, showing how complex the infection process can be.

**Table 1 ijms-21-09634-t001:** Description of secreted RNAs by different bacterial species.

Bacterial Species	Sequenced Biological Samples	seRNAs Features	Host Cell	Remarks	References
*Aggregatibacter actinomycetemcomitans* (ATCC 33384)	Outer OMV-associated secreted vs. cell small RNAs.	ncRNAs sizing 15 to 28 nt, some derived from degraded products, ribosomal RNA (rRNA) and transfer RNA (tRNA)	Jurkat T-cells Human macrophage-like cells (U937)	msRNA A.A_20050 reduce anti-inflammatory cytokines Interleukin (IL)–5, IL-13, and IL-15 in vitro. Cytoplasmic delivery and activity of microbial EV-derived small seRNAs in macrophages. Enhanced TNF-α production via TLR-8 and NF-κB signalling pathways.	[103,105]
*Borrelia burgdorferi* B31	Outer OMV-associated secreted vs. cell RNA.	Reduced content of rRNA. The majority of small ncRNAs found on OMVs is derived from chromosomes.	N/A	Most of small ncRNAs are enriched in OMVs. Enrichment on plasmid-encoded transcripts in the OMVs, suggesting an involvement in genomic rearrangement	[111]
*Chlamydia trachomatis* LGV-L2 434/Bu	Secreted small RNAs inside host cells vs. cell small RNAs.	ncRNA of 15 to 43 nt size. Among the ten most abundant bacterial small RNAs 9 were originated of ORF regions and just one from non-coding regions.	HeLa CCL-2 cells		[101]
*Escherichia coli* 536	Outer OMV-associated secreted vs. cell RNA.	rRNA, tRNAs, other small RNAs. Undegraded mRNAs. Abundant small size RNA (15–50 nt)	5637 bladder epithelial cells	1% of MV RNA cargo delivered into cultured cells, including sRNA csrC	[87]
*Escherichia coli* K-12 substrain MG1655	Outer free and OMV-associated secreted vs. cell RNA (<200 nt)	Selective exportation of 15 to 40 nt long cleavage products of tRNAs, 4.5S RNA, 6S RNA and tmRNA.	N/A		[77]
*Helicobacter pylori* J99	Outer OMV-associated secreted vs. cell RNA.	59 ncRNAs present in OMVs. 45 ncRNAs enriched in OMVs.	Human gastric adenocarcinoma cells	seRNAs are delivered to host cells. seRNAs sR-2509025 and sR-989262 reduce interleukin 8 (IL-8) secretion	[79]
*Legionella pneumophila* CR39	Secreted small RNAs inside host cells vs. cell small RNAs.	ncRNA of 15 to 43 nt size. Among the ten most abundant bacterial small RNAs 7 were originated from tRNA and 3 from ORF regions.	Murine macrophage cell line RAW264.7		[101]
*Listeria monocytogenes* EGD-e	Outer free and OMV-associated secreted vs. cell RNA.	seRNAs are mainly of short size (<200 nt), especially in OMVs	BMDM, P388D1 macrophages, and HEK293 cells	seRNAs highly induce IFN-β. ncRNA rli32-induced IFN- β in a RIG-I dependent way, leading to an inhibition of influenza virus replication. rli32 are involved in *L. monocytogenes* grow ability inside host cells. The RBP Zea bind seRNAs and RIG-I.	[80,98]
*Mycobacterium marinum* M	Secreted small RNAs inside host cells vs. cell small RNAs.	ncRNA of 15 to 43 nt size. Among the ten most abundant bacterial small RNAs 9 were originated of ORF regions and just one from non-coding regions.	Murine macrophage cell line RAW264.7	The small RNA MM-H presented features of a miRNA was able to repress the host mRNA-target MM-H-specific RLuc reporter.	[101]
*Mycobacterium smegmatis*	Secreted small RNAs inside host cells vs. cell small RNAs.	ncRNA of 15 to 43 nt size. The ten most abundant bacterial small RNAs were originated from coding regions.	Murine macrophage cell line RAW264.7		[101,109]
*Mycobacterium tuberculosis*	Secreted small RNAs inside host cells vs. cell small RNAs.	ncRNA of 15 to 43 nt size. Among the ten most abundant bacterial small RNAs 9 were originated of ORF regions and just one from non-coding regions.	Murine macrophage cell line RAW264.7	seRNA compromise human monocytes’ ability to control infection. seRNA secretion through sec2A secretory system. EVs released from *M. tuberculosis* infected macrophages enhance host defences.	[81,101,102]
non-typeable *Haemophilus influenzae*	N/A	RNA present in biofilm matrix	N/A	Extracellular RNA appears to be required for the initial attachment of a biofilm	[112]
*Porphyromonas gingivalis* (ATCC 33277)	Outer OMV-associated secreted vs. cell small RNAs.	ncRNAs sizing 15 to 50 nt, some derived from degraded products, ribosomal RNA (rRNA) and transfer RNA (tRNA). mRNAs of virulence factors detected by qRT-PCR	Jurkat T-cells, human gingival fibroblasts, human oral keratinocytes and human umbilical vein endothelial cells	msRNA P.G_45033 reduce anti-inflammatory cytokines Interleukin (IL)–5, IL-13, and IL-15 in vitro. OMVs can penetrate host cells. Vesicle-mediated RNA transference between *P. gingivalis* strains.	[103,104]
*Pseudomonas aeruginosa* PA14	Outer OMV-associated secreted vs. cell RNA.	64 ncRNAs present in OMVs. 52 ncRNAs enriched in OMVs.	Human bronchial epithelial (HBE) cells	sRNA52320 reaches HBE cells, reduces IL-8 secretion by cultured HBE cells and attenuate KC cytokine secretion and neutrophil infiltration in mouse lung	[74]
*Pseudomonas aeruginosa* PAO1	Outer MV-associated secreted vs. cell RNA.	Intact 16S and 23S rRNAs. mRNAs related to SOS response (oxidative stress) enriched in cell lysis derived MVs	N/A	Reduced small RNA PA3305.1 (PhrS) content in MVs, suggesting an explosive cell lysis mediated MV production independent of PQS.	[89]
*Salmonella enterica* serovar Typhimurium LT2	Outer OMV-associated secreted vs. cell RNA.	Mainly rRNA. mRNA and ncRNA were specifically enriched in OMVs from different growth conditions.	N/A	RNA is protected from degradation in OMVS. RNA cargo in OMVs is dependent of growth conditions and content abundance differs from the cell	[69]
*Salmonella enteritidis* SE2472	Secreted RNA inside host cells vs. cell RNA.			RNA secretion inside host cells. ncRNA Sal-1 assume its mature form by host cell Argonaute 2 (AGO2) processing. Sal-1 increases infection rate in mice and the cell survival rate of *Salmonella*	[99]
*Streptococcus* *pyogenes* (GAS)	Outer OMV-associated secreted vs. cell RNA.	Mainly mRNAs, rRNAs and tRNAs	N/A	Different from most gram-negative bacteria, GAS membrane vesicles is enriched in intragenic RNA	[110]
*Streptococcus sanguinis*	N/A	msRNA detected in OMVs by qRT-PCR	N/A	msRNA are found in inside bacteria and are secreted through OMVs	[106]
*Treponema denticola* (ATCC 35405)	Outer OMV-associated secreted vs. cell small RNAs.	ncRNAs sizing 15 to 28 nt, some derived from degraded products, ribosomal RNA (rRNA) and transfer RNA (tRNA)	Jurkat T-cells	msRNA T.D_2161 reduce anti-inflammatory cytokines Interleukin (IL)–5, IL-13, and IL-15 in vitro	[103]
*Vibrio cholerae* A1552	Outer OMV-associated secreted vs. cell RNA.	Abundant content of ncRNAs	N/A	RNA from the bacterial cells in association with OMVs are not growth phase dependent	[107]

Abbreviations: N/A, not available

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
