# Peer review of "Extracellular RNAs in Bacterial Infections: From Emerging Key Players on Host-Pathogen Interactions to Exploitable Biomarkers and Therapeutic Targets"

_ijms, 2020, doi:10.3390/ijms21249634_

Round 1

Reviewer 1 Report

Dear Editor,

I carefully read the submission entitled “Extracellular RNAs in Bacterial Infections: from emerging key players on host-pathogen interactions to exploitable biomarkers and therapeutic targets” by Pita and colleagues.

In this review, authors present the RNA secretion as a ubiquitous strategy to communicate. In particular in bacteria this secretion occurs mainly through vesicles that provides protection from RNase degradation.

In the introduction, the different types of non-coding RNAs are presented in both eukaryotes and prokaryotes. In the second paragraph a clear explanation of the five mechanisms used to export non-coding RNAs to extracellular environment was expected.

Moreover, the examples reported about the process of loading of ncRNAs are related to cancer and not to bacterial infections.

The third paragraph should be one of the most important in this review but, in my opinion, it is not well presented. It is not clear, an explicative image is missing and the last part from line 231 there is simply a list of published works. Also the last paragraph needs major attention and clear conclusions of the state of art should emerge since that this aspect is also in the title of the manuscript and thus of importance for readers.

Minor comments:

-          Line 97 replace eukarya with eukaryote in the paragraph title.

-          Line 121 citation in number

-          Line 157 citation 39 write …by Kumar et al. 39….

-          In page 6 correct many citations reporting numbers close to authors’ names ex line 237, 244, 247,250,256,263, 287,309,402,436,437…

Because of concerns above reported I discourage the publication of this review.

Author Response

RESPONSES TO Reviewer #1

COMMENT: I carefully read the submission entitled “Extracellular RNAs in Bacterial Infections: from emerging key players on host-pathogen interactions to exploitable biomarkers and therapeutic targets” by Pita and colleagues.

In this review, authors present the RNA secretion as a ubiquitous strategy to communicate. In particular in bacteria this secretion occurs mainly through vesicles that provides protection from RNase degradation.

In the introduction, the different types of non-coding RNAs are presented in both eukaryotes and prokaryotes. In the second paragraph a clear explanation of the five mechanisms used to export non-coding RNAs to extracellular environment was expected.

ANSWER: Thanks for the suggestion. We have introduced an explanation of the 5 mechanisms used to export non-coding RNAs to the extracellular environment. See new lines 108-153, which now reads as follows: At least five mechanisms of miRNAs release to the extracellular environment have been described: i) miRNA bound to RNA-protein complexes. A couple of studies have identified the secretion of miRNAs linked to proteins of the Agonaut family, like Agonaut 2 (Ago2), a type of proteins that are associated to RNA-inducing silencing complex (RISC) 28. RISC is involved in most of miRNAs regulation 29. In fact, the majority of miRNAs found in human plasma are bounded to Agonaut proteins, but this seems to be mainly related to cell death and not to a selective secretion of miRNAs 30; ii) transport via lipid or lipoprotein particles. In addition to be crucial transporters of steroids, triglycerols, cholesterol and fat-soluble vitamins, low density lipoproteins (LDL) and mainly high density lipoproteins (HDL) can also play a role in miRNAs intercellular communication 31,32. The loading mechanism seems to involve divalent cation bridging between miRNAs and HDL 31; inside microvesicles. Microvesicles are formed by plasma membrane by budding or fission, and therefore, its lipid content are quite similar to the parent cell membrane 33. Microvesicles can be a way of secretion of many types of molecules, including nucleic acids, being also responsible for the cell-to-cell communication by miRNAs exportation in several clinical conditions 34. Although the evidences of the evolvement of microvesicles miRNAs on intracellular communication, their relevance in infection conditions remains unknown, as well as are still unclear the sorting and loading processes 35; iv) inside vesicles from apoptotic bodies. Apoptosis is a natural process of controlled cell death by eukaryotic cells. In this process, the release of apoptotic bodies, the greatest vesicles secreted by eukaryotic cells, is common 36. Like microvesicles, its content can be vast, including miRNAs, mRNAs and DNA fragments, but it can also be selective, and under specific conditions some miRNAs can be highly represented on these vesicles 37. Again, the sorting mechanism remains unknown 35; v) inside exosomes. Despite their origin or structure, extracellular vesicles (EV) seem to be the most usual mechanism to selectively export ncRNAs to the extracellular space 27. Exosomes seem to play a special role on cell-to-cell communication on infections conditions 35. Exosomes are generated inside endosomes or multivesicular bodies (MVBs), and released through fusion of these exosome-enriched late endosomes with the plasma membrane (Figure 1) 38. Although the process of sorting and loading of ncRNAs is still poorly characterized, is about exosomes that we know the most. In animals, the heterogeneous nuclear ribonucleoprotein A2B1 (hnRNPA2B1) was described to control the exosomal loading of miRNAs by binding to specific “EXOmotifs” on these miRNAs 39. The most detailed example found in literature is about colorectal cancer cells. In this case, the KRAS-MEK signalling pathway seems to be responsible for the regulation of exosomal loading of the RISC component Argonaute 2, an RNA-binding protein and a key effector of miRNA-guided RNA silencing process 40. In addition, during the sorting and loading of ncRNAs in exosomes, other RNA-binding proteins are also suggested to participate, like the Y-box protein required for the miR-233 secretion by human embryonic kidney (HEK)293T cells 41, and the SYNCRIP protein required for miRNA sorting in hepatocytes. miRNAs found in those exosomes possess an extra-seed sequence (hEXO motif) that binds to these RNA-binding proteins 42. Still in the hepatic system, the RNA-binding protein Vps4A was found to mediate the flux of miRNAs through exosomes. Vps4A facilitates the secretion of oncogenic miRNAs in exosomes, while promoting the accumulation and uptake of tumour suppressor miRNAs in cells. A downregulation of this protein was observed in hepatocellular carcinoma (HCC) cells 43. In addition, the transcriptional regulation of miRNAs expression or their targets also implies a miRNA sorting regulation on exosome secretion, as it has been shown in macrophages and endothelial cells communication 44. The mechanism of sorting and loading of miRNAs in infections conditions remain to be investigated.

COMMENT: Moreover, the examples reported about the process of loading of ncRNAs are related to cancer and not to bacterial infections.

ANSWER: Thanks for the comment. As examples of ncRNAs loading related to bacterial infections remain to be found, we decided to keep the descriptions. In addition, this is now stated in new lines 152-153, which now reads as follows:

The mechanism of sorting and loading of miRNAs in infections conditions remain to be investigated

COMMENT: The third paragraph should be one of the most important in this review but, in my opinion, it is not well presented. It is not clear, an explicative image is missing and the last part from line 231 there is simply a list of published works.

ANSWER: Thanks for the comment and suggestion. In order to accommodate the various criticisms, previous points #3 and #4 were merged into the new point 3: Extracellular ncRNAs in Bacteria. Therefore, the suggestion to include a new Figure was not followed, as Figure 1 already illustrates the content described in the text.

COMMENT:Also the last paragraph needs major attention and clear conclusions of the state of art should emerge since that this aspect is also in the title of the manuscript and thus of importance for readers.

ANSWER: Thanks for the suggestion. In order to reinforce this aspect new information was added to the manuscript. Please see new lines 533-542 and 561-563, which now reads as follows:

Lines 533-542

Fecal miRNAs are also promising as a new way of diagnosis. The discovery of a different pattern  of secreted miRNAs  in conditions like colorectal cancer and celiac disease, as well as the use of miRNAs to modulated gut microbiome metabolism, has paved the way for the development of new biomarkers based on these secreted molecules 117. On the animal field, a total of 14 secreted miRNAs were found to be differentially expressed in bovine milk exosomes in response to a Staphylococcus aureus infection in mammary gland118. This example evidence that a selective secretion of ncRNAs can occur in an infection condition, and biomarkers of infections can be originated from different sources, by the infectious agent itself, or by host cells, as a mechanism of defence. The potential of using seRNAs as infections biomarker exists, however none real application has become available so far.

Lines 561-563:

Also, Miravirsen, the first miRNA-targeted drug, is still under clinical validation, currently in Phase II clinical trial 117. There is a growing hope in the use of this kind of therapies, but the challenges associated with safety and efficiency are huge 122.

Reviewer 2 Report

This is a very good review in an area of host-pathogen interactions that remains under explored. The manuscript is well structured and written. An excellent standard of English is presented but the authors do need a thorough read-through for typos and grammatical errors.

The review gives a few examples of bacterial infections and extracellular RNAs but this is rather brief. I feel this needs to be expanded to include the latest papers on this subject for the bacterial pathogens described. I used PubMed and found numerous recent papers that are not cited. Furthermore, I think more detail is needed on the biological mechanisms and consequences in bacterial infection. This is touched upon, but its rather brief. what are the sequence composition and similarities of different extracellular RNAs from bacterial species?

Lastly, the article's title indicates the possible use of these RNA species as biomarkers and therapeutic agents. These are not fully explored in the article. The phrase 'Scratching' (line 466) is not helpful and should be replaced and this section should be expanded as suggested above. 

However, Excellent figures and table presented.

Overall, a very good review, that is rather brief on specific detail and content.

Author Response

RESPONSES TO REVIEWER #2

COMMENT: This is a very good review in an area of host-pathogen interactions that remains under explored. The manuscript is well structured and written. An excellent standard of English is presented but the authors do need a thorough read-through for typos and grammatical errors.

The review gives a few examples of bacterial infections and extracellular RNAs but this is rather brief. I feel this needs to be expanded to include the latest papers on this subject for the bacterial pathogens described. I used PubMed and found numerous recent papers that are not cited. Furthermore, I think more detail is needed on the biological mechanisms and consequences in bacterial infection. This is touched upon, but its rather brief. what are the sequence composition and similarities of different extracellular RNAs from bacterial species?

ANSWER: Thank you for you kind appreciation and constructive criticism. Although there is a relatively abundant literature on bacterial sRNAs and pathogenesis, literature on bacterial EXTRACELLULAR sRNAs and pathogenesis is quite limited. Nevertheless, we have performed additional search using Pubmed and a few additional examples not included in the previous version were added, in particular in point 3.3.8. In addition, we have merged the previous sections #3 and #4, to focus on bacterial extracellular sRNAs and additional information was added to the specific sections on bacterial pathogens. Since the modifications are extensive, please refer to the manuscript with marked modifications were the changes are highlighted in grey. As a whole, we consider that the new section 3 greatly improved thanks to your suggestions.

COMMENT: Lastly, the article's title indicates the possible use of these RNA species as biomarkers and therapeutic agents. These are not fully explored in the article. The phrase 'Scratching' (line 466) is not helpful and should be replaced and this section should be expanded as suggested above. 

ANSWER: Thanks for the criticism. This section was revised and additional information was added, see new lines 533-542 and 561-563, already shown above as part of the response to a similar comment from reviewer#1.

Round 2

Reviewer 1 Report

Dear Editor,

The authors partially addressed my concerns. In my opinion the paper does not full meet the high level of the journal. Anyway  if other reviewers do not share my concerns I will support an editorial decision of 'accept'.

Reviewer 2 Report

I am satisfied with the changes made by the authors in light of my review.